# Learning English in Early Childhood Education with Augmented Reality: Design, Production, and Evaluation of the "Wordtastic Kids" App



Aleyda Mamani-Calapuja [1], Victoria Laura-Revilla [1], Alejandra Hurtado-Mazeyra [2] and Carmen Llorente-Cejudo [3,*]

1. Departamento de Educación Programa de Idiomas, Universidad Nacional de San Agustín de Arequipa, Santa Catalina 117, Peru; amamanical@unsa.edu.pe (A.M.-C.); vlaurar@unsa.edu.pe (V.L.-R.)
2. Academic Department of Education, Universidad Nacional de San Agustín de Arequipa, Santa Catalina 117, Peru; ahurtadomaz@unsa.edu.pe
3. Department of Teaching and Educational Organization, University of Sevilla, Pirotecnia Street, 41013 Seville, Spain
* Correspondence: karen@us.es

**Abstract:** The pedagogical use of AR for language learning in early childhood education is focusing attention on the didactic potential that these resources present at such early ages. The aim of this study was to develop and assess the "Wordtastic Kids" Application (APP) in order to: (a) design, produce, and evaluate a new contextualized AR application for learning English in early childhood education; and (b) know the academic performance of early childhood education students towards learning English using an AR application. The methodology was based on the design of materials and the subsequent realization of the pre-test/post-test evaluation to determine the academic performance of the students. The results show a general increase in the scores obtained in all the pre-test and post-test categories that were addressed using the Wordtastic Kids Application, which proves to be a tool that improves the learning of English vocabulary in pre-school children. Therefore, it can be concluded that AR can be useful for complementing traditional vocabulary learning in children. However, it is important to note that the successful implementation of AR will depend on several factors, such as adequate teacher preparation, the quality of AR content, and the ability of children to interact with technology effectively.

**Keywords:** educational applications; early childhood education; augmented reality; English learning

## 1. Introduction

Augmented Reality (AR) is an emerging technology focused on projecting digital 3D images onto reality [1]; that is, spaces become new environments where virtual objects are incorporated and take part in reality in different ways [2], with a connection between these two worlds, thus creating a combined reality that requires the use of a technological device. AR can be considered the projection of an image on a surface that allows interacting with or recognizing objects as well as adding information about them [3,4]. Among the benefits of using AR in education are:

- Provides contextual learning;
- Promotes the constructive ideas of education;
- Fosters the creativity and imagination of the students;
- Prevents the real consequences of making mistakes during the training of skills;
- Stimulates the students and motivates them to explore;
- Improves teacher-student collaboration.

Several studies affirm that the use of augmented reality presents a significant efficacy in the achievement of learning due to its interactivity characteristics highlighting

students' acceptance, comfort, engagement, and self-efficacy, but with greater significance, the findings evidence its potential in motivation and the results of knowledge acquisition (including memorization, retention, and application) [5,6], thus facilitating visualization and interaction with information and promoting the development of cognitive and motor skills [7,8]. Research on augmented reality has focused on teaching people to use this technology effectively in the learning environments it generates and in its educational application [9].

However, there are few studies on the AR-app teaching of English in early childhood education. We developed the present study in line with previous works, such as that of [10], who also pointed out that teaching English in early childhood education has become an important aspect of the development of computerization and economic globalization.

## 2. Theoretical Framework: Augmented Reality

*Augmented Reality Applications for Learning English in Education*

Several studies have focused their interest on the pedagogical use of AR for the learning of languages, such as that of [11], which analyzed whether the use of AR improves the learning of English as a foreign language, increases student motivation, and helps children establish more positive socio-emotional relationships. Their findings demonstrate a significant improvement in motivation, socio-emotional relationships, and the learning of English in the group of students who used AR in the classroom (the experimental group) compared to those who followed the traditional method (the control group). Applications such as Aurasma and Quiver were used, which allowed the students to interact with AR and emphasize the potential of an immersive context. Similarly, the study [12] focused on the effective use of AR to improve the learning experience of early childhood education students; this was achieved through a mobile application prototype with AR to teach English vocabulary interactively, attractively, and at any time, using a mobile device with a monitoring system in the app. The preliminary evaluation showed that the effectiveness of the app was satisfactory, thereby demonstrating that AR could be used in early childhood education, so long as the time of use is mediated and controlled. In the same study, two mobile app prototypes with AR were developed to teach English vocabulary in early childhood. The first application allows learning anytime and anywhere using a mobile device, and it has a monitoring device that allows stopping the app remotely. The second app allows teachers to use AR to plan their teaching as well as improve the quality of their learning experiences. The results show a satisfactory perception of the use of both applications for the teaching of English, and the authors propose increasing the number of materials, the number of activities, and the use of more graphic buttons, as well as improving the acceptance of the monitoring system.

Based on the situational theory that teaches words and evaluates the progress at different ages in early childhood education [10], we proposed a new method to teach English through the design of AR resources, which may help the app achieve excellent effectiveness. Through the use of software to scan AR cards, children can watch scenes in 3D and 3D models of the words, and they can also listen to their pronunciation. The results show that the group of children who underwent the AR experience (the experimental group) presented better results in the learning of vocabulary, with variations in the different ages in the number of words memorized, and they also remembered a larger number of words compared to those who followed the traditional method (the control group). In [13], educational experiences using AR technology were used to improve the acquisition of vocabulary and grammatical structures in English, introducing the curricular contents of emotional intelligence. They incorporated activities with AR to promote autonomous learning through exploration and self-evaluation, as well as images and audio materials to facilitate the learning of the contents, including phonetics through narrations and songs. The evaluation that was conducted in six classrooms shows a very positive acceptance of the methodology by the students, demonstrating positive effects on learning outcomes and high rates of motivation and participation. In this line [14], we developed an application

with AR called "AR-TO-KID", whose aim was to improve the pronunciation of words in English, characterized by voice entry detection in children of early childhood education aged 5–6 years. In this sense, the application requires the child to give an answer based on his/her reasoning by understanding the AR objects that appear after digitizing the trigger. The app consists of three different scenarios, which contain three 3D images that represent the possible answers. The correct answer appears in 3D only when the user manages to pronounce the correct alternative with a British accent. A pre-test and a post-test were applied to observe the improvement in the learning of the English language, along with an instrument to measure the satisfaction of the students. The results confirm that the application improved the pronunciation problems as well as the critical thinking of the children. Moreover, it was observed that most of the children had fun using the application, thereby increasing their participation and maintaining their attention in the activities for a longer time.

In order to measure the impact of the "AURASMA RA" App [15], we carried out a study using tablets. To this end, they worked with two groups of children in the first year of primary education: the experimental group worked with the AR application, and the control group used cards, following the traditional method. The results report that the students of the experimental group showed a significant improvement in their motivation for learning the English language, whereas the control group did not present any differences. Regarding learning performance, the results of the experimental group were more significant. The authors concluded that both methods stimulated all three senses analyzed (sight, touch, and kinesthesia). The traditional method stimulates hearing and sight through singing and teaching with cards, whereas teaching with AR stimulates hearing and sight through 3D images and promotes kinesthesia through interaction with the screen and peers.

In this respect [16], we established the efficacy of using an AR application with cards for acquiring a second language in a population of 52 children from a kindergarten who were divided into an experimental group and a control group. The results revealed that the use of AR was effective in the children who used it, showing a significantly greater performance both on the same day and in the 3-day delayed productive recognition, with better results in intrinsic motivation while learning vocabulary compared to the control group [17]. We evaluated the impact of AR technology in early childhood education through two methods: one with the use of AR cards of animals and the other with traditional paper cards. A total of 98 children aged 5–6 years participated in the study, and the experiment was carried out with four teachers. To measure the effectiveness of the two approaches, the children were subjected to pre- and post-tests on their vocabulary, and the teachers were interviewed using Student's t-tests for paired samples and independent samples to measure the effect size. The results showed that both AR and the traditional cards significantly improved the vocabulary of the children and that there were no differences in effectiveness between AR and the traditional method. The teachers stated that the children enjoyed the AR learning, although there were certain challenges associated with the use of AR flashcards in an environment of early childhood education [18]. conducted a study using AR technology for the interactive learning of phonetics in English in consonant-vowel-consonant (CVC) words in early childhood education. A test was applied to measure whether the didactic material proposed was interactive and attractive for the children. A total of 30 children from two kindergartens participated in the study. The children were required to read 15 CVC words before and after using the didactic material proposed. The technique based on AR image markers allowed the children to interact with virtual phonetic content through physical handling. The results demonstrate that the children learned the phonetic sound with 3D letters by interacting with a single phonetic card. The tests and the evaluation showed that the didactic material was effective, precise, and efficient.

The results of the performance evaluation also show that the children improved in reading after using this didactic material. In this sense, [19] reported the effectiveness of an AR app called TeachAR, which combines AR and voice recognition, in the learning

of English words for colors, shapes, and spatial associations. This system is the first AR language learning tool designed for teaching young children, aged 4–6 years, about spatial associations and 3D shapes. A total of 120 children from six different kindergartens in Malaysia participated in the study. They were divided into eight groups, with fifteen children in each group. The results show that the interaction in real-time increased the motivation of the children to further explore the learning materials. All participants found it easy to interact with the system. The authors concluded that the AR interface with voice recognition can be used with young children with little or no experience; moreover, it allows the user to complete certain tasks faster and more easily. As an environmental drawback, it is difficult to control the noise level in the classroom while the voice-activated mode of TeachAR is used, and since strong noise can reduce the performance of voice recognition, the children may feel frustrated.

From this perspective, we proposed the development, analysis, and implementation of the "Wordtastic Kids" app with the aim of:

1. Designing, producing, and evaluating a new AR app contextualized for learning English in early childhood education;
2. Determining the degree of motivation of students in early childhood education towards AR and learning English;
3. Knowing the academic performance of students in early childhood education about learning English using an AR app.

## 3. Methodology

To attain the objectives of our study, different phases were carried out, which are presented in Figure 1 below.

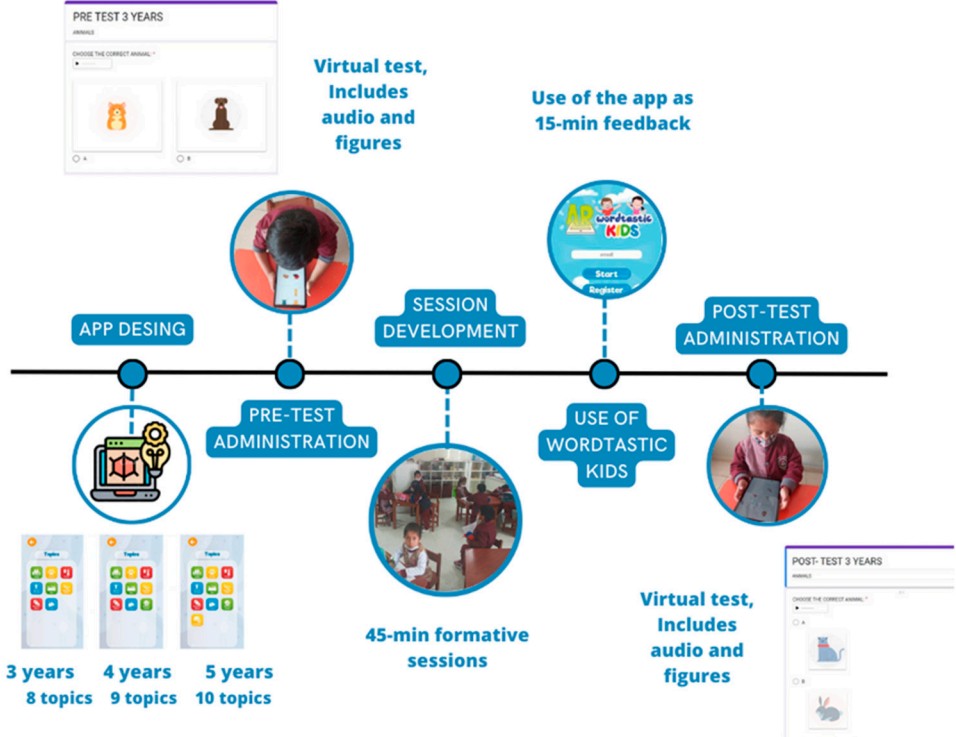

**Figure 1.** Scheme of the study phases.

### 3.1. Design of the Application

The Wordtastic Kids App was designed by a research team of specialists in education, early childhood education, and languages. Based on an Android system, its design is simple and intuitive for use by early childhood students through animations and the pronunciation of the words that correspond to each object, which promotes the motivation

of the students to learn English vocabulary. Its use requires the trigger or marker shown below (Figure 2), which is a single QR code that activates the animations in AR.

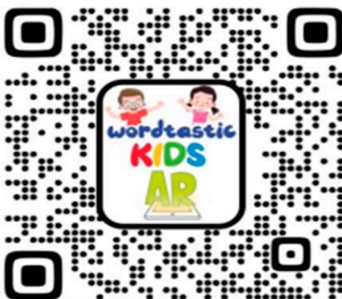

**Figure 2.** Trigger/marker of the app.

Next, Figure 3 shows how to operate the Wordtastic Kids App.

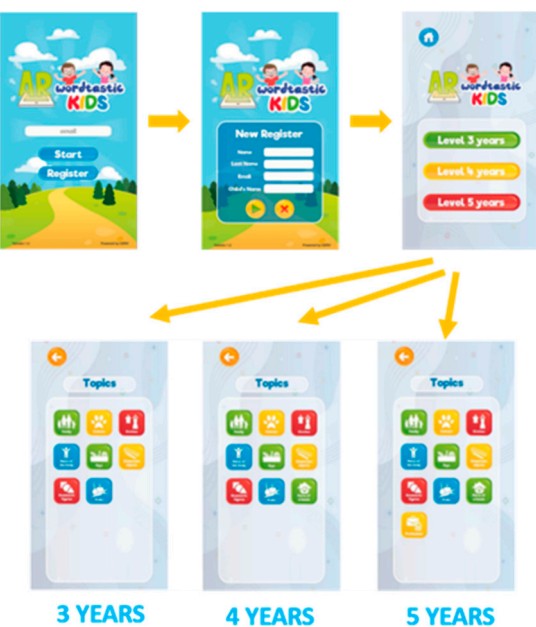

**Figure 3.** A sequence of images showing how to operate the app.

Upon initiating the app, the logo appears on the screen, and then the user registers with an email address. Subsequently, a main menu comes up, where the user can select the level according to age (i.e., level 3 years, level 4 years, level 5 years), based on real topics that are constantly worked on in each level. In addition to the options, there is a house icon, which is used to exit the application.

By touching one of the options, a new menu with topics is opened, where the user selects the vocabulary to work on. These topics include family, animals (i.e., pets), clothing, parts of the body, toys, classroom objects, geometric shapes, and fruits (for level 3 years). For level 4 years, the topics are similar, although the topic of animals is broader, and this level also includes a new topic that corresponds to parts of the household. In level 5 years, the topic of animals is expanded to include jungle animals, and a new category of professions is included.

To activate the AR objects, the user must place the camera of the tablet on the trigger, which will allow him/her to see the object that corresponds to the category. Likewise, the user can interact with the AR objects in the following manner:

4. By pressing on the object, an audio track in English is reproduced, along with the animation that corresponds to it;

5.    Using two fingers on the object, the latter can be expanded or reduced;
6.    A horizontal movement of the object makes it rotate.

When the user wishes to continue with the next object, he/she only needs to touch the arrow to proceed, and when the last object of the topic is reached, an end-of-session icon appears along with an animation to congratulate the user for his/her achievement.

### 3.2. Participants

Children aged 3–5 years from a Peruvian kindergarten participated in the study. The participants were recruited by non-probabilistic convenience sampling, with an initial sample of 69 students. However, an inclusion criterion was regular attendance by the children due to the duration of the program and the distribution of English topics. The final sample was constituted by 42 children, who were distributed in the following manner according to age: 3 years (12 children), 4 years (13 children), and 5 years (17 children). In the Figure 4 shows how the children used the "Wordtastic Kids" app.

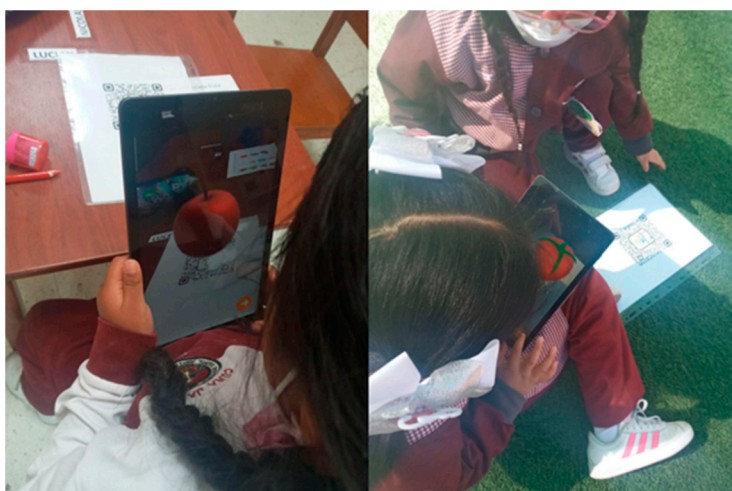

**Figure 4.** Students interacting with the "Wordtastic Kids" app.

We obtained the informed consent form of the parents about the voluntary participation of their children and the procedure throughout the investigation, following the ethical guidelines.

### 3.3. Instrument

To measure the vocabulary level of the children, the research team designed and developed an instrument through Google Forms, which was made available for use on tablets. The instrument is differentiated by age and by the topics included in the app. The user registers before and after the session, where a specific vocabulary topic is developed (family, animals, clothing, etc.). The maximum score is 20 points. The instrument was validated through a pilot test, which allowed us to establish the completion time, corresponding to an average of 4–5 min per topic. The audio and figures included in the app allowed the children to register autonomously, as can be observed in Figure 5.

### 3.4. Procedure

The educational intervention with the AR application was conducted with three groups of children: 3, 4, and 5 years of age. Before starting the intervention, a chronogram was created for the distribution of the sessions by topics and ages concerning the application within the school hours assigned for English classes. Following that, the formative sessions were designed, diversifying them by age. Subsequently, the pre-test was applied, as described in the previous section. After that, the formative sessions were developed, which included the use of specific and representative material about the planned topic and had

an approximate duration of 45 min. For the closing of the session, the Wordtastic Kids Application was used as a feedback tool, using tablets and explaining to each child how it works. The approximate time of use for this phase was 15 min on average, and it was conducted in small groups of 3–4 students. The day after the development of each session, we applied the post-test to evaluate the learning of vocabulary as a function of the topic addressed and verify the efficacy of the application. The same process was followed until all the topics included in the app were completed.

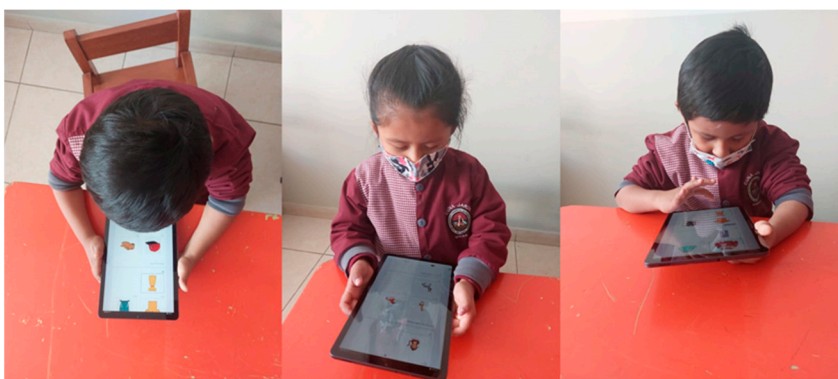

**Figure 5.** Students completing the pre-test and post-test.

*3.5. Data Analysis*

The data were treated using the statistical package SPSS v21 for Windows (Statistical Package for Social Sciences). The data analysis includes descriptive statistics, frequencies, percentages, and means of the answers given by the students in the questionnaires. Likewise, non-parametric tests were used to carry out the comparative analysis about possible significant differences in the variables included in the questionnaire as a function of age, gender, and previous knowledge of AR tools.

## 4. Results

Next, we present the obtained results as a function of the general descriptive statistics found in the pre-test and post-test.

As can be observed in Table 1, there was a general increase in the scores obtained in all categories of pre-test and post-test that were studied through the use of the Wordtastic Kids App, such as "Animals", with a mean score increase of 10 to 18.7; "Body", with 8.8 to 16.8; and "Clothes", with the greatest pre-test/post-test difference (7.4 to 16.1). Moreover, the standard deviation scores were always higher in the pre-test and lower in the post-test in all the analyzed categories.

**Table 1.** Means and standard deviations of the analyzed categories in the pre-test and post-test, before and after implementing the Wordtastic Kids App, respectively.

| Categories | N | | Mean | Standard Deviation |
| --- | --- | --- | --- | --- |
| | Valid | Missing | | |
| Animals_pre | 42 | 0 | 10.0952 | 4.57890 |
| Animals_pos | 42 | 0 | 18.7143 | 2.14460 |
| Body_pre | 42 | 0 | 8.8571 | 3.80354 |
| Body_pos | 42 | 0 | 16.8095 | 2.60617 |
| Classroom_pre | 42 | 0 | 10.7857 | 4.24572 |
| Classroom_pos | 42 | 0 | 16.6429 | 2.78341 |
| Clothes_pre | 42 | 0 | 7.4524 | 3.52830 |
| Clothes_pos | 42 | 0 | 16.1429 | 3.23526 |

**Table 1.** *Cont.*

| Categories | N | | Mean | Standard Deviation |
|---|---|---|---|---|
| | Valid | Missing | | |
| Family_pre | 42 | 0 | 10.6429 | 3.99891 |
| Family_pos | 42 | 0 | 17.3333 | 2.94392 |
| Fruit_pre | 42 | 0 | 13.2143 | 4.89667 |
| Fruit_pos | 42 | 0 | 18.1190 | 2.86444 |
| Geometric_pre | 42 | 0 | 13.4286 | 4.89969 |
| Geometric_pos | 42 | 0 | 19.0000 | 2.10690 |
| Toys_pre | 42 | 0 | 15.4762 | 3.14866 |
| Toys_pos | 42 | 0 | 18.6667 | 1.99593 |

The results of the general descriptive statistics by age are presented in Table 2.

**Table 2.** Means and standard deviations of the analyzed categories in the pre-test and post-test, before and after implementing the Wordtastic Kids App, respectively, by age.

| Categories | 3 Years | | 4 Years | | 5 Years | |
|---|---|---|---|---|---|---|
| | Mean | Standard Deviation | Mean | Standard Deviation | Mean | Standard Deviation |
| Animals_pre | 10.42 | 4.98 | 8.38 | 4.61 | 11.18 | 4.13 |
| Animals_pos | 18.75 | 2.26 | 17.46 | 2.70 | 19.65 | 0.79 |
| Body_pre | 9.58 | 3.45 | 9.54 | 3.76 | 7.82 | 4.05 |
| Body_pos | 16.58 | 2.27 | 17.08 | 3.01 | 16.76 | 2.63 |
| Classroom_pre | 10.42 | 3.48 | 10.08 | 4.68 | 11.59 | 4.50 |
| Classroom_pos | 16.67 | 1.56 | 16.85 | 3.02 | 16.47 | 3.36 |
| Clothes_pre | 8.00 | 4.51 | 7.23 | 2.24 | 7.24 | 3.72 |
| Clothes_pos | 16.33 | 2.67 | 15.15 | 3.74 | 16.76 | 3.19 |
| Family_pre | 12.00 | 5.39 | 10.69 | 2.93 | 9.65 | 3.48 |
| Family_pos | 18.00 | 2.09 | 18.46 | 2.30 | 16.00 | 3.46 |
| Fruit_pre | 12.50 | 4.52 | 12.38 | 5.82 | 14.35 | 4.43 |
| Fruit_pos | 17.50 | 3.37 | 18.69 | 1.80 | 18.12 | 3.20 |
| Geometric_pre | 13.83 | 6.59 | 11.69 | 3.45 | 14.47 | 4.36 |
| Geometric_pos | 20.00 | 0.00 | 18.77 | 2.52 | 18.47 | 2.35 |
| Toys_pre | 14.67 | 3.11 | 15.92 | 3.75 | 15.71 | 2.73 |
| Toys_pos | 17.33 | 2.61 | 19.08 | 1.89 | 19.29 | 0.99 |

As can be observed, all age groups showed an increase in the means between the pre-test and the post-test after the implementation of the instrument. Moreover, the pre-test/post-test differences were similar for the different age groups regarding the analyzed categories: in the 3-year group, the categories with a greater increase were Animals (10.42 to 18.75; 8 increment points) and Clothes (8 to 16.33); in the 4-year group, the greatest increases were found in Animals (11.18 to 19.65), Clothes (7.23 to 15.15), and Family (10.69 to 18.46); and, in the 5-year group, the categories with the most significant increase were Animals (11.18 to 19.65), Clothes (7.24 to 16.76), and Body (7.82 to 16.76).

Likewise, it is worth highlighting that most of the categories with the lowest increase also coincide among groups: in the 3-year group, the categories with the lowest increase were Toys (14.67 to 17.33), Fruit (12.50 to 17.50), and Classroom (10.42 to 16.67); in the

4-year group, the lowest increases were found in Toys (15.92 to 19.08), Fruit (12.38 to 18.69), and Classroom (10.08 to 16.85); and, in the 5-year group, the categories with the lowest increase were Toys (15.71 to 19.29), Fruit (14.35 to 18.12), and Geometric Shapes (14.47 to 18.47).

Therefore, according to age, these results indicate that the use of the Wordtastic Kids App in the acquisition of English vocabulary through AR involved an increase in both the pre-test and the post-test of all the analyzed categories that make up the app, with such an increase taking place in the same categories and all age groups.

Regarding the analysis of the obtained data, to compare the pre-test and post-test results in a general manner, the Wilcoxon signed-rank test was performed as shown in Table 3.

**Table 3.** General statistics of the Wilcoxon signed-rank test.

| | Test Statistics [a] | |
| --- | --- | --- |
| | **Z** | **Asymptotic Sig. (Bilateral)** |
| Animals_pos—Animals_pre | −5.520 [b] | 0.000 |
| Body_pos—Body_pre | −5.584 [b] | 0.000 |
| Classroom_pos—Classroom_pre | −5.519 [b] | 0.000 |
| Clothes_pos—Clothes_pre | −5.587 [b] | 0.000 |
| Family_pos—Family_pre | −5.398 [b] | 0.000 |
| Fruit_pos—Fruit_pre | −4.552 [b] | 0.000 |
| Geometric_pos—Geometric_pre | −5.044 [b] | 0.000 |
| Toys_pos—Toys_pre | −4.578 [b] | 0.000 |

[a] Wilcoxon signed-rank test. [b] Based on negative ranks.

All analyzed cases showed significant differences between the pre-test and post-test scores ($p < 0.05$). To verify which test (pre-test or post-test) these differences tended to occur in, a mean rank analysis was performed (Table 4), which shows that the positive ranks always obtained greater scores. As was previously mentioned, it is evident that, in all cases, the differences appeared in the post-test, and the mean rank indicates that the highest scores were obtained in the post-test.

**Table 4.** Analysis of mean ranks and sum of ranks.

| | | Ranks | | |
| --- | --- | --- | --- | --- |
| | | **N** | **Mean Rank** | **Rum of Ranks** |
| Animals_pos—Animals_pre | Negative ranks | 0 [a] | 0.00 | 0.00 |
| | Positive ranks | 40 [b] | 20.50 | 820.00 |
| | Ties | 2 [c] | | |
| | Total | 42 | | |
| Body_pos—Body_pre | Negative ranks | 0 [d] | 0.00 | 0.00 |
| | Positive ranks | 41 [e] | 21.00 | 861.00 |
| | Ties | 1 [f] | | |
| | Total | 42 | | |
| Classroom_pos—Classroom_pre | Negative ranks | 0 [g] | 0.00 | 0.00 |
| | Positive ranks | 40 [h] | 20.50 | 820.00 |
| | Ties | 2 [i] | | |
| | Total | 42 | | |

**Table 4.** *Cont.*

| | | N | Mean Rank | Rum of Ranks |
|---|---|---|---|---|
| | | **Ranks** | | |
| | | **N** | **Mean Rank** | **Rum of Ranks** |
| Clothes_pos—Clothes_pre | Negative ranks | 0 [j] | 0.00 | 0.00 |
| | Positive ranks | 41 [k] | 21.00 | 861.00 |
| | Ties | 1 [l] | | |
| | Total | 42 | | |
| Family_pos—Family_pre | Negative ranks | 0 [m] | 0.00 | 0.00 |
| | Positive ranks | 38 [n] | 19.50 | 741.00 |
| | Ties | 4 [o] | | |
| | Total | 42 | | |
| Fruit_pos—Fruit_pre | Negative ranks | 0 [p] | 0.00 | 0.00 |
| | Positive ranks | 27 [q] | 14.00 | 378.00 |
| | Ties | 15 [r] | | |
| | Total | 42 | | |
| Geometric_pos—Geometric_pre | Negative ranks | 1 [s] | 3.50 | 3.50 |
| | Positive ranks | 33 [t] | 17.92 | 591.50 |
| | Ties | 8 [u] | | |
| | Total | 42 | | |
| Toys_pos—Toys_pre | Negative ranks | 0 [v] | 0.00 | 0.00 |
| | Positive ranks | 27 [w] | 14.00 | 378.00 |
| | Ties | 15 [x] | | |
| | Total | 42 | | |

Note: Average range superscripts: a. Animals_pos < Animals_pre; b. Animals_pos > Animals_pre; c. Animals_pos = Animals_pre; d. Body_pos < Body_pre; e. Body_pos > Body_pre; f. Body_pos = Body_pre; g. Classroom_pos < Classroom_pre; h. Classroom_pos > Classroom_pre; i. Classroom_pos = Classroom_pre; j. Clothes_pos < Clothes_pre; k. Clothes_pos > Clothes_pre; l. Clothes_pos = Clothes_pre; m. Family_pos < Family_pre; n. Family_pos > Family_pre; o. Family_pos = Family_pre; p. Fruit_pos < Fruit_pre; q. Fruit_pos > Fruit_pre; r. Fruit_pos = Fruit_pre; s. Geometric_pos < Geometric_pre; t. Geometric_pos > Geometric_pre; u. Geometric_pos = Geometric_pre; v. Toys_pos < Toys_pre; w. Toys_pos > Toys_pre; x. Toys_pos = Toys_pre

Applied also to each of the age groups, we can observe (Tables 5 and 6 for the 3-year group, Tables 7 and 8 for the 4-year group, and Tables 9 and 10 for the 5-year group) that significance remained positive in each of the analyzed categories ($p < 0.005$).

**Table 5.** General statistics of the Wilcoxon signed-rank test in the 3-year group in the pre-test/post-test.

| | Z | Asymptotic Sig. (Bilateral) |
|---|---|---|
| | **Test Statistics** [a] | |
| | **Z** | **Asymptotic Sig. (Bilateral)** |
| Animals_pos—Animals_pre | −2.976 [b] | 0.003 |
| Body_pos—Body_pre | −2.938 [b] | 0.003 |
| Classroom_pos—Classroom_pre | −3.088 [b] | 0.002 |
| Clothes_pos—Clothes_pre | −2.956 [b] | 0.003 |
| Family_pos—Family_pre | −2.871 [b] | 0.004 |
| Fruit_pos—Fruit_pre | −2.460 [b] | 0.014 |
| Geometric_pos—Geometric_pre | −2.414 [b] | 0.016 |
| Toys_pos—Toys_pre | −2.530 [b] | 0.011 |

[a] Wilcoxon signed-rank test. [b] Based on negative ranks.

**Table 6.** Analysis of the mean ranks and sum of ranks in the 3-year group in the pre-test/post-test.

| | | N | Mean Rank | Sum of Ranks |
|---|---|---|---|---|
| | | **Ranks** | | |
| | | N | Mean Rank | Sum of Ranks |
| Animals_pos—Animals_pre | Negative ranks | 0 [a] | 0.00 | 0.00 |
| | Positive ranks | 11 [b] | 6.00 | 66.00 |
| | Ties | 1 [c] | | |
| | Total | 12 | | |
| Body_pos—Body_pre | Negative ranks | 0 [d] | 0.00 | 0.00 |
| | Positive ranks | 11 [e] | 6.00 | 66.00 |
| | Ties | 1 [f] | | |
| | Total | 12 | | |
| Classroom_pos—Classroom_pre | Negative ranks | 0 [g] | 0.00 | 0.00 |
| | Positive ranks | 12 [h] | 6.50 | 78.00 |
| | Ties | 0 [i] | | |
| | Total | 12 | | |
| Clothes_pos—Clothes_pre | Negative ranks | 0 [j] | 0.00 | 0.00 |
| | Positive ranks | 11 [k] | 6.00 | 66.00 |
| | Ties | 1 [l] | | |
| | Total | 12 | | |
| Family_pos—Family_pre | Negative ranks | 0 [m] | 0.00 | 0.00 |
| | Positive ranks | 10 [n] | 5.50 | 55.00 |
| | Ties | 2 [o] | | |
| | Total | 12 | | |
| Fruit_pos—Fruit_pre | Negative ranks | 0 [p] | 0.00 | 0.00 |
| | Positive ranks | 7 [q] | 4.00 | 28.00 |
| | Ties | 5 [r] | | |
| | Total | 12 | | |
| Geometric_pos—Geometric_pre | Negative ranks | 0 [s] | 0.00 | 0.00 |
| | Positive ranks | 7 [t] | 4.00 | 28.00 |
| | Ties | 5 [u] | | |
| | Total | 12 | | |
| Toys_pos—Toys_pre | Negative ranks | 0 [v] | 0.00 | 0.00 |
| | Positive ranks | 7 [w] | 4.00 | 28.00 |
| | Ties | 5 [x] | | |
| | Total | 12 | | |

Note: Average range superscripts: a. Animals_pos < Animals_pre; b. Animals_pos > Animals_pre; c. Animals_pos = Animals_pre; d. Body_pos < Body_pre; e. Body_pos > Body_pre; f. Body_pos = Body_pre; g. Classroom_pos < Classroom_pre; h. Classroom_pos > Classroom_pre; i. Classroom_pos = Classroom_pre; j. Clothes_pos < Clothes_pre; k. Clothes_pos > Clothes_pre; l. Clothes_pos = Clothes_pre; m. Family_pos < Family_pre; n. Family_pos > Family_pre; o. Family_pos = Family_pre; p. Fruit_pos < Fruit_pre; q. Fruit_pos > Fruit_pre; r. Fruit_pos = Fruit_pre; s. Geometric_pos < Geometric_pre; t. Geometric_pos > Geometric_pre; u. Geometric_pos = Geometric_pre; v. Toys_pos < Toys_pre; w. Toys_pos > Toys_pre; x. Toys_pos = Toys_pre.

**Table 7.** General statistics of the Wilcoxon signed-rank test in the 4-year group in the pre-test/post-test.

| | Test Statistics [a] | |
|---|---|---|
| | **Z** | **Asymptotic Sig. (Bilateral)** |
| Animals_pos—Animals_pre | −3.200 [b] | 0.001 |
| Body_pos—Body_pre | −3.193 [b] | 0.001 |
| Classroom_pos—Classroom_pre | −3.187 [b] | 0.001 |
| Clothes_pos—Clothes_pre | −3.187 [b] | 0.001 |
| Family_pos—Family_pre | −3.194 [b] | 0.001 |
| Fruit_pos—Fruit_pre | −2.677 [b] | 0.007 |
| Geometric_pos—Geometric_pre | −3.236 [b] | 0.001 |
| Toys_pos—Toys_pre | −2.588 [b] | 0.010 |

[a] Wilcoxon signed-rank test. [b] Based on negative ranks.

**Table 8.** Analysis of mean ranks and sum of ranks in the 4-year group in the pre-test/post-test.

| | Ranks | | | |
|---|---|---|---|---|
| | | **N** | **Mean Rank** | **Sum of Ranks** |
| Animals_pos—Animals_pre | Negative ranks | 0 [a] | 0.00 | 0.00 |
| | Positive ranks | 13 [b] | 7.00 | 91.00 |
| | Ties | 0 [c] | | |
| | Total | 13 | | |
| Body_pos—Body_pre | Negative ranks | 0 [d] | 0.00 | 0.00 |
| | Positive ranks | 13 [e] | 7.00 | 91.00 |
| | Ties | 0 [f] | | |
| | Total | 13 | | |
| Classroom_pos—Classroom_pre | Negative ranks | 0 [g] | 0.00 | 0.00 |
| | Positive ranks | 13 [h] | 7.00 | 91.00 |
| | Ties | 0 [i] | | |
| | Total | 13 | | |
| Clothes_pos—Clothes_pre | Negative ranks | 0 [j] | 0.00 | 0.00 |
| | Positive ranks | 13 [k] | 7.00 | 91.00 |
| | Ties | 0 [l] | | |
| | Total | 13 | | |
| Family_pos—Family_pre | Negative ranks | 0 [m] | 0.00 | 0.00 |
| | Positive ranks | 13 [n] | 7.00 | 91.00 |
| | Ties | 0 [o] | | |
| | Total | 13 | | |
| Fruit_pos—Fruit_pre | Negative ranks | 0 [p] | 0.00 | 0.00 |
| | Positive ranks | 9 [q] | 5.00 | 45.00 |
| | Ties | 4 [r] | | |
| | Total | 13 | | |

**Table 8.** *Cont.*

| | | N | Mean Rank | Sum of Ranks |
|---|---|---|---|---|
| **Ranks** | | | | |
| | Negative ranks | 0 [s] | 0.00 | 0.00 |
| Geometric_pos—Geometric_pre | Positive ranks | 13 [t] | 7.00 | 91.00 |
| | Ties | 0 [u] | | |
| | Total | 13 | | |
| | Negative ranks | 0 [v] | 0.00 | 0.00 |
| Toys_pos—Toys_pre | Positive ranks | 8 [w] | 4.50 | 36.00 |
| | Ties | 5 [x] | | |
| | Total | 13 | | |

Note: Average range superscripts: a. Animals_pos < Animals_pre; b. Animals_pos > Animals_pre; c. Animals_pos = Animals_pre; d. Body_pos < Body_pre; e. Body_pos > Body_pre; f. Body_pos = Body_pre; g. Classroom_pos < Classroom_pre; h. Classroom_pos > Classroom_pre; i. Classroom_pos = Classroom_pre; j. Clothes_pos < Clothes_pre; k. Clothes_pos > Clothes_pre; l. Clothes_pos = Clothes_pre; m. Family_pos < Family_pre; n. Family_pos > Family_pre; o. Family_pos = Family_pre; p. Fruit_pos < Fruit_pre; q. Fruit_pos > Fruit_pre; r. Fruit_pos = Fruit_pre; s. Geometric_pos < Geometric_pre; t. Geometric_pos > Geometric_pre; u. Geometric_pos = Geometric_pre; v. Toys_pos < Toys_pre; w. Toys_pos > Toys_pre; x. Toys_pos = Toys_pre

**Table 9.** General statistics of the Wilcoxon signed-rank test in the 5-year group in the pre-test/post-test.

| | Z | Asymptotic Sig. (Bilateral) |
|---|---|---|
| **Test Statistics [a]** | | |
| Animals_pos—Animals_pre | −3.522 [b] | 0.000 |
| Body_pos—Body_pre | −3.628 [b] | 0.000 |
| Classroom_pos—Classroom_pre | −3.416 [b] | 0.001 |
| Clothes_pos—Clothes_pre | −3.628 [b] | 0.000 |
| Family_pos—Family_pre | −3.423 [b] | 0.001 |
| Fruit_pos—Fruit_pre | −2.949 [b] | 0.003 |
| Geometric_pos—Geometric_pre | −3.088 [b] | 0.002 |
| Toys_pos—Toys_pre | −3.074 [b] | 0.002 |

[a] Wilcoxon signed-rank test. [b] Based on negative ranks.

**Table 10.** Analysis of the mean ranks and sum of ranks in the 5-year group in the pre-test/post-test.

| | | N | Mean Rank | Sum of Ranks |
|---|---|---|---|---|
| **Ranks** | | | | |
| | Negative ranks | 0 [a] | 0.00 | 0.00 |
| Animals_pos—Animals_pre | Positive ranks | 16 [b] | 8.50 | 136.00 |
| | Ties | 1 [c] | | |
| | Total | 17 | | |
| | Negative ranks | 0 [d] | 0.00 | 0.00 |
| Body_pos—Body_pre | Positive ranks | 17 [e] | 9.00 | 153.00 |
| | Ties | 0 [f] | | |
| | Total | 17 | | |

**Table 10.** *Cont.*

| | | **Ranks** | | |
|---|---|---|---|---|
| | | **N** | **Mean Rank** | **Sum of Ranks** |
| Classroom_pos—Classroom_pre | Negative ranks | 0 [g] | 0.00 | 0.00 |
| | Positive ranks | 15 [h] | 8.00 | 120.00 |
| | Ties | 2 [i] | | |
| | Total | 17 | | |
| Clothes_pos—Clothes_pre | Negative ranks | 0 [j] | 0.00 | 0.00 |
| | Positive ranks | 17 [k] | 9.00 | 153.00 |
| | Ties | 0 [l] | | |
| | Total | 17 | | |
| Family_pos—Family_pre | Negative ranks | 0 [m] | 0.00 | 0.00 |
| | Positive ranks | 15 [n] | 8.00 | 120.00 |
| | Ties | 2 [o] | | |
| | Total | 17 | | |
| Fruit_pos—Fruit_pre | Negative ranks | 0 [p] | 0.00 | 0.00 |
| | Positive ranks | 11 [q] | 6.00 | 66.00 |
| | Ties | 6 [r] | | |
| | Total | 17 | | |
| Geometric_pos—Geometric_pre | Negative ranks | 1 [s] | 3.50 | 3.50 |
| | Positive ranks | 13 [t] | 7.81 | 101.50 |
| | Ties | 3 [u] | | |
| | Total | 17 | | |
| Toys_pos—Toys_pre | Negative ranks | 0 [v] | 0.00 | 0.00 |
| | Positive ranks | 12 [w] | 6.50 | 78.00 |
| | Ties | 5 [x] | | |
| | Total | 17 | | |

Note: Average range superscripts: a. Animals_pos < Animals_pre; b. Animals_pos > Animals_pre; c. Animals_pos = Animals_pre; d. Body_pos < Body_pre; e. Body_pos > Body_pre;f. Body_pos = Body_pre; g. Classroom_pos < Classroom_pre; h. Classroom_pos > Classroom_pre; i. Classroom_pos = Classroom_pre; j. Clothes_pos < Clothes_pre; k. Clothes_pos > Clothes_pre; l. Clothes_pos = Clothes_pre; m. Family_pos < Family_pre; n. Family_pos > Family_pre; o. Family_pos = Family_pre; p. Fruit_pos < Fruit_pre; q. Fruit_pos > Fruit_pre; r. Fruit_pos = Fruit_pre; s. Geometric_pos < Geometric_pre; t. Geometric_pos > Geometric_pre; u. Geometric_pos = Geometric_pre; v. Toys_pos < Toys_pre; w. Toys_pos > Toys_pre; x. Toys_pos = Toys_pre

Furthermore, to determine the existence of significant differences between the scores obtained by the different age groups and the types of tests, the Kruskal-Wallis test was carried out as a non-parametric method. In this sense, none of the differences (Tables 11 and 12) were significant since none of the values obtained were higher than 0.005. Therefore, these values allow us to assert that the scores of the students in the pre-test and post-test are homogeneous across the analyzed age groups (3, 4, and 5 years).

**Table 11.** Kruskal-Wallis in the grouping variable "age" for the pre-test.

| | **Animals_Pre** | **Body_Pre** | **Classroom_Pre** | **Clothes_Pre** | **Family_Pre** | **Fruit_Pre** | **Geometric_Pre** | **Toys_Pre** |
|---|---|---|---|---|---|---|---|---|
| | | | | **Test Statistics** | | | | |
| Kruskal-Wallis H | 2.692 | 3.227 | 0.634 | 0.435 | 3.270 | 2.001 | 4.144 | 2.255 |
| df | 2 | 2 | 2 | 2 | 2 | 2 | 2 | 2 |
| Asymptotic sig. | 0.260 | 0.199 | 0.728 | 0.805 | 0.195 | 0.368 | 0.126 | 0.324 |

**Table 12.** Kruskal-Wallis in the grouping variable "age" in the post-test.

| | Animals _Pos | Body _Pos | Classroom _Pos | Clothes _Pos | Family_Pos | Fruit _Pos | Geometric _Pos | Toys _Pos |
|---|---|---|---|---|---|---|---|---|
| | | | | **Test Statistics** | | | | |
| Kruskal-Wallis H | 8.047 | 0.215 | 0.405 | 1.541 | 5.471 | 0.512 | 4.783 | 6.004 |
| df | 2 | 2 | 2 | 2 | 2 | 2 | 2 | 2 |
| Asymptotic sig. | 0.018 | 0.898 | 0.817 | 0.463 | 0.065 | 0.774 | 0.091 | 0.050 |

## 5. Discussion and Conclusions

The "Wordtastic Kids" App is a tool that allows for improving the learning of English vocabulary in children in early childhood education. In this study, a learning experience was carried out with children in a kindergarten using AR. The instrument was designed and developed with a series of images, sounds, and commands that facilitate learning interactively.

While it is true that in the educational institution where Augmented Reality was applied, it was observed that it has an English area for the levels of 3, 4, and 5 years old. Prior to the implementation of our sessions and the application, information was collected from the teachers through general questions to determine the degree of familiarity of the children with the vocabulary to be addressed. However, it was indicated that some topics, such as family and animals, had not yet been worked on, while others, such as classroom objects, had only been addressed in a limited way. In addition, it was noted that many children, when moving from one level to another, tend to forget some of the previously acquired vocabulary. It was also identified that some children were new to the school and had never had contact with the language, which generated the satisfaction of knowing that our work would contribute to reinforcing and expanding their vocabulary.

The results of this study were obtained thanks to the planning and execution of the learning sessions addressed by the researchers and the application of AR as a complement to the learning experience, which allowed students to learn in a motivating and interesting way. The scores obtained by all age groups on all vocabulary items in the post-test after using the "Wordtastic Kids" App provided an interactive and immersive experience by combining virtual elements with additional real-time information such as flash cards, among others. The greatest differences were obtained in the topics about animals and clothes for the 3-year group, in the topics about animals, clothes, and family for the 4-year group, and in the topics of animals, clothes, and parts of the body for the 5-year group. On the other hand, it is worth mentioning that there was a lower increase in topics about toys, fruits, classroom objects, and geometric shapes in all age groups.

Therefore, the positive results obtained in this study are in line with our general aim of exploring the viability of the use of AR in teaching English to children in early childhood education, considering that the children are not in a bilingual context and learn English as a second language since all the teachers are Peruvian; hence, the exposure of children to the English language is predominantly limited to their school environment, where they engage solely during classroom sessions with their teacher. Within these sessions, various activities such as crafting, solving exercises, and other interactive tasks are employed to facilitate language learning. Consequently, the need to assign homework is deemed unnecessary. These results are supported by previous studies, such as that of [10,13,14,16], who reported that the preschool children who used AR acquired English more efficiently compared to the group who used the conventional approach, which entails the use of standard materials and a teacher-centered instructional model where information is imparted in a directive manner. Within this framework, students typically assume a passive role characterized by obedience, focusing on memorization and repetition instead of actively engaging with three-dimensional (3D) materials. The limited integration of 3D materials restricts students' opportunities to actively participate, think critically, and apply their knowledge. On their part [15], they also obtained favorable results in English learning through the use of AR, although their study was conducted on primary education students. In both studies, the immersive experience provided by AR was highlighted as a

key to success, implying m-learning, or mobile learning, in which mobile devices such as smartphones and tablets are highlighted as a valuable means of enhancing the learning process. Offering numerous advantages, including portability, connectivity, multimedia capabilities, and tactile functionality, they are used to create interactive, attractive, and fun experiences that encourage learning.

Thus, we can assert that AR is an interactive tool that can significantly improve the process of learning the English language for students in early childhood education by integrating AR into educational activities. Different characteristics of AR can improve the hearing, visual, and pronunciation skills of children aged 3, 4, and 5, which enriches their learning experience. Moreover, we must consider that the children who participated in this study are not familiar with this type of technology and that they only use traditional didactic materials such as flashcards, songs, and videos in their daily classes. AR allows students to interact with virtual objects and situations in real-time, which helps them develop their creativity, critical thinking, and problem-solving skills by provoking their interest and motivation to use something new by manipulating the object themselves.

Another study conducted by [20] shows that the positive results observed in the learning of preschool children through virtual 3D models are due to their novelty and visual attractiveness, which allow them to interact with an environment that resembles reality. Moreover, since this technology is little known to preschool children, they become curious about it, which increases their motivation and concentration. This application is focused on preschool learning, and thus it is easy to use, which significantly improves its effectiveness, as the technology employed in it is simple and allows the user to explore and understand the educational content.

Similarly, other studies agree on using technology as a teaching-learning strategy, for addressing an innovative approach that improves interaction, understanding, motivation, and personalization of learning and that taking into account the design and objective proposed in the session, educators can take advantage of the benefits of AR and offer more attractive and effective learning experiences for students, for instance, the study of [21] shows that thanks to the flexibility and adaptability of AR, it is possible to improve the formative processes in different environments highlighting that immersive experiences can capture the interest and attention of students; by merging the real world with virtual elements; an environment is created that stimulates curiosity and the desire to explore. Likewise, the study [22] highlights the use of technology in the educational context, where AR becomes an effective vehicle for learning. Using it in regular classroom lessons, technology offers flexibility and accessibility to learning where students can access educational materials anytime and anywhere through mobile devices, stimulating participation and collaborating in their learning process. After using it in classes taught by teachers of English as a second language (ESL), English as a foreign language (EFL), and bilingual education (BE), positive attitudes were observed in the students, such as greater participation and a more effective learning process.

Therefore, it can be concluded that AR can be a useful tool to complement the traditional learning [23,24] of vocabulary by children. However, it is important to take into account that the successful implementation of AR will depend on several factors, such as the adequate preparation of the teacher, the quality of the AR content, and the capacity of the children to interact with technology effectively; considering that there are possible disadvantages to be taken into account, especially at the age of kindergarten children, such as sensory overload consisting of too many simultaneous visual or auditory stimuli, which could be overwhelming and confusing for children; dependence on technology involving difficulty in social interaction; then it is important to balance the use of technology with other activities and pedagogical approaches so as not to replace the actual experience of touching, feeling and experimenting with physical objects.

For future work, it is important for researchers to consider both the advantages and disadvantages of using technology in education and how it can be effectively implemented.

In addition, relevant information should be provided so that educational institutions and teachers understand the purpose and objectives of using technology in the classroom.

It is essential to consider students' needs and carefully plan the use of technology applications to ensure that their attention is captured and effective learning is achieved in a didactic manner. This requires teachers to be adequately prepared, familiar with the use of the applications, and able to integrate them coherently and effectively into their educational activities.

In addition, it is necessary to conduct periodic evaluations of the use of technological applications to ensure that they contribute to the learning process and to avoid overexposure to technology, which can negatively impact children.

**Author Contributions:** Methodology, A.M.-C., V.L.-R. and A.H.-M.; validation, A.H.-M., C.L.-C.; formal analysis, C.L.-C.; investigation, A.M.-C., V.L.-R. and A.H.-M.; visualization, A.M.-C., V.L.-R.; supervision, A.M.-C., V.L.-R. and A.H.-M.; project administration, A.M.-C. All the authors equally contributed to the conceptualization, writing—review and editing and writing—original draft preparation. All authors have read and agreed to the published version of the manuscript.

**Funding:** This research was funded by UNIVERSIDAD NACIONAL DE SAN AGUSTÍN DE ARE-QUIPA contract N° TP CS-05-2020-UNSA.

**Institutional Review Board Statement:** The study did not require ethical approval.

**Informed Consent Statement:** Informed consent was obtained from all subjects involved in the study.

**Data Availability Statement:** The relevant data is primarily contained within the article.

**Acknowledgments:** The authors thank the UNSA Initial Education Institution for their support in the project.

**Conflicts of Interest:** The authors declare no conflict of interest.

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
