# Peer review of "Learning English in Early Childhood Education with Augmented Reality: Design, Production, and Evaluation of the “Wordtastic Kids” App"

_education, doi:10.3390/educsci13070638_

Round 1
Reviewer 1 Report
Very interesting study but some questions come to my mind, that I think they haven't been mentioned in the text and could improve it.
1. In conclusions it is stated that "an increase in the scores obtained by all age groups in 321 all the vocabulary topics of the post-test after using the “Wordtastic Kids” App."... It could be due to the exposition to the language, working on certain topics, maybe or not thanks to the app.
How do we know that the improvement is merely due to the use of Wordtastic Kids app? Please expand on it.
2. How familiar were the students with the topics worked on: family, fruits, classroom... as those are corte topics already practiced in class at that age.
3. The context. What kind of early language learning are they working on? Is it a bilingual context, how exposed are they to English? In school, at home... as that will really affect attainment.
4. Line 339. How are kinesthetic skills promoted using AR. No previous mentioning in the article and is asserted in the conclusions.
5. Line 362. Mentioning "traditional learning", meaning without 3D materials? There is no explanation in the context on which methodology or practices are being carried out with the students. Are they familiar with the use of technology in English class or is it included in their routines? Is it something new thus attractive and motivating?
6. Line 365. Were the teachers part of the implementation of the app? Meaning, did they have any specific training to do so?
7. Lines 368-376. Needs more in depth and focused conclusions. Expand this paragraph (section).
Thank you
Author Response
We incorporate attached with all the modifications to the suggestions that were indicated to us in the review.
Thank you

Reviewer 2 Report
This article complies well with the essential elements of quality, although certain aspects need to be improved.
Firstly, the number of up-to-date, international (and high-impact) references on Augmented Reality should be increased.
Secondly, a significant effort should be made to improve the discussion and conclusions. There is no strong comparison or contrast with previous studies, and what is presented does not manage to coherently scrutinise all the didactic, formal and curricular aspects that can be addressed in this section.
Thirdly, the focus is more technological than didactic, and a more inclusive and open perspective is required in order to project the image of usability, transferability and innovation in the use of these applications to improve the quality of education in all its terms and facets.
Author Response
We incorporate all the modifications in an attached file.
Thank you

Reviewer 3 Report
Lines 66 to 80 should include a commentary on the visual and interactive potential provided by the use of AR in the educational context. The studies https://doi.org/10.1344/der.2021.39.%25p and https://doi.org/10.6018/educatio.427011 have shown that the use of AR has an attractive and effective educational potential as long as the underlying didactic proposal is well structured and appropriate to the educational needs of the students. That is why we recommend including both articles to support the idea of this paragraph.
Lines 362 to 366. The reflection raised is interesting. In order to achieve an effective analysis of the App's potential, it would be useful to outline the possible difficulties and setbacks faced by teachers who opt for the use of this app, as well as the details of using AR with students of such a young age.
Author Response

(The authors gave the same response as above.)
